# Shiga Toxin-Producing *E. coli* (STEC) from Danish Patients, 1997–2023: Diagnostic Trends and Bacteriological Findings

**DOI:** 10.3390/microorganisms13102342

**Published:** 2025-10-12

**Authors:** Flemming Scheutz, Katrine Grimstrup Joensen, Susanne Schjørring, Bente Olesen, Jørgen Engberg, Hanne Marie Holt, Hans Linde Nielsen, Lars Lemming, Michael Pedersen, Lisbeth Lützen, Marc Trunjer Kusk Nielsen, Kristian Schønning, Eva Møller Nielsen

**Affiliations:** 1The International *Escherichia* and *Klebsiella* Centre, Section of Foodborne Infections, Department of Bacteria, Parasites and Fungi, Statens Serum Institute, 2300 Copenhagen, Denmark; knj@ssi.dk (K.G.J.); ssc@ssi.dk (S.S.); emn@ssi.dk (E.M.N.); 2Department of Clinical Microbiology, Copenhagen University, Herlev Hospital, Herlev and Gentofte, Borgmester Ib Juuls Vej 52, 2730 Herlev, Denmark; bente.ruth.scharvik.olesen@regionh.dk; 3Department of Clinical Microbiology, Zealand University Hospital, 4600 Koege, Denmark; jheg@regionsjaelland.dk; 4Department of Clinical Microbiology, Odense University Hospital, 5000 Odense, Denmark; hanne.holt@rsyd.dk; 5Department of Clinical Microbiology, Aalborg University Hospital, 9000 Aalborg, Denmark; 6Department of Clinical Medicine, Aalborg University, 9220 Aalborg, Denmark; 7Department of Clinical Microbiology, Aarhus University Hospital, 8200 Aarhus, Denmark; larslemm@rm.dk; 8Department of Clinical Microbiology, Section 445, Copenhagen University Hospital, Amager/Hvidovre, Kettegaard Alle 30, 2650 Hvidovre, Denmark; michael.pedersen.06@regionh.dk; 9Department of Clinical Microbiology, Sygehus Lillebælt, 6000 Vejle, Denmark; lisbeth.lutzen@rsyd.dk; 10Department of Clinical Diagnostics, Syddansk Universitetshospital, Esbjerg og Grindsted Sygehus, Finsensgade 35, 6700 Esbjerg, Denmark; marc.nielsen@rsyd.dk; 11Department of Clinical Microbiology, Copenhagen University Hospital—Rigshospitalet, 2100 Copenhagen, Denmark; kristian.schoenning@regionh.dk

**Keywords:** Shiga toxin-producing *Escherichia coli* (STEC), surveillance, molecular detection, whole genome sequencing (WGS), cross over strains, syndromic testing, haemolytic uraemic syndrome (HUS), diagnostic trends

## Abstract

Implementation of molecular detection methodology of Shiga toxin-producing *Escherichia coli* (STEC) in Danish patients began in 1997. Since then, changes in molecular detection methods and diagnostic criteria have led to the present situation, in which almost all diarrhoeal stool specimens are examined for STEC. Whole genome sequencing (WGS) of STEC isolates referred to the national reference laboratory has increased the detailed characterisation, and revealed a large spectrum, of STEC types, including cross-over pathotypes typically associated with extraintestinal disease or traveller’s diarrhoea. Association of subtype *stx2a* (and *stx2d*) with the risk of developing haemolytic uraemic syndrome (HUS) was confirmed. These changes have resulted in an increase in the number of diagnosed STEC cases from 31 cases in 1997 to 1432 in 2023. Similar increases in Europe have also been recorded. Culture of STEC is, on the other hand, declining, which poses a challenge to the identification of multiple STEC infections and outbreaks. Syndromic (PCR) test panels have also resulted in an increase in the detection of multiple microorganisms. Double or triple infections have increased the role of clinical microbiologists in interpreting and assessing the significance of diagnostic results and have also increased the need for high-quality curation of surveillance data.

## 1. Introduction

Shiga toxin-producing *Escherichia coli* (STEC), previously referred to as Verocytotoxin-producing *E. coli* (VTEC), is considered as one of the most important human pathogens in developed countries [1]. Globally, STEC are estimated to cause 2.8 million illnesses annually, leading to approximately 3890 cases of haemolytic uraemic syndrome (HUS) and 230 deaths [2]. STEC infections are the fourth most reported gastrointestinal foodborne illness in humans in the European Union, with 8565 and 10,217 confirmed cases, 568 and 892 HUS cases, and 28 and 31 deaths in 2022 and 2023, respectively [3,4]. STEC has caused foodborne outbreaks in Denmark and abroad [5,6,7,8,9]. STEC O157 is considered one of the most costly foodborne diseases [10]. However, the epidemiology of human infections caused by STEC has changed over the past decades with non-O157 STEC now playing an increasingly important role. It has previously been found that infections caused by non-O157 STEC strains have been underdiagnosed [11,12,13]. In Denmark, only a few cases of STEC were recorded prior to 1996.

During the study period from 1997 to 2023, some major changes have taken place: traditional detection methods have been replaced by either in-house real-time (RT-) PCR or commercial gastrointestinal panels, clinical criteria have been changed from selected stools to examination of all diarrhoeal specimens for STEC, and whole genome sequencing (WGS) has been routinely applied since 2015. STEC and clinical HUS became notifiable in 2000. As similar trends have been seen worldwide, we therefore wanted to give an overview of the consequences of these changes in the surveillance of STEC.

In this study, we describe the characterisation of the Danish STEC collected as part of national surveillance from 1997 to 2023.

## 2. Materials and Methods

### 2.1. Study Period and Population

In 1997, the department of gastrointestinal infections at SSI, receiving stool cultures from 12 DCMs representing 77% of the Danish population, changed diagnostic practices: bloody stools, stools with a history of bloody diarrhoea, stools from persistent diarrhoea, i.e., diarrhoea for more than two weeks, and from travellers were all examined for the most common diarrhoeagenic *E. coli* (DEC). All stools from children younger than four years were examined for STEC during the first three years (1997–1999), and from children younger than seven years from 2000 to 2015. From 2016 to 2023, SSI tested all referred specimens regardless of clinical criteria. During the study period, primary detection changed as a consequence of the decentralisation of primary diagnostic practices to regional hospitals using traditional culture methods and their own diagnostic criteria. By the end of 1999, only seven DCMs representing 49% of the Danish population were covered by the SSI diagnostic practice. A detailed overview of the methods, clinical criteria, and year of introduction and changes are shown in the T1 in the Appendix A (SM).

### 2.2. Primary Diagnostic Methods

Until 2003, the primary diagnostic methods used at SSI were a combination of dot blot hybridisation using pooled DNA probes and live slide agglutination of suspect colonies with O-antisera against the most common STEC and EPEC (enteropathogenic *E. coli*) serotypes. From 2003, detection was changed to include a DEC PCR as described previously [14].

The exact methods and procedures, including additional key references, are described in detail in the Appendix A (SM).

### 2.3. Detection and Characterisation at Departments of Clinical Microbiology

Regional hospitals initially diagnosed STEC by live slide agglutination of suspect colonies with a panel of OK antisera, typically O groups O26, O103, O111, O145 and O157. In 2003, the Danish Society for Clinical Microbiology (DSCM) adopted guidelines for the detection of STEC and even though regional differences existed, the majority of regional hospitals followed the recommendations by the DSCM by expanding their diagnostic practice to include different DNA methods such as the DEC PCR, the octaplex PCR [15], in-house Q-PCR, and commercially available Shiga toxin gene (*stx*) detection kits such as QIAstat-Dx* Gastrointestinal Panel, BD MAX, or Seegene. The use of diagnostic methods and indications to test for DEC in the period from 1997 to 2023 at SSI and in 11 DCMs is shown in T1 in Appendix A.

### 2.4. Characterisation at the National Reference Laboratory at SSI

Until 2021, selected faecal specimens and all STEC cultures were routinely submitted to the National Reference Laboratory (NRL) at Statens Serum Institut (SSI) for further characterisation, allowing for outbreak detection and a comparison of incidence, prevalence of specific types, and association between types and specific clinical features in relation to differences in diagnostic approaches.

Since 2022, attempts to isolate STEC have primarily been limited towards *stx2a* and/or *stx2d* PCR-positive specimens and on suspicion of outbreak. These isolates are then referred to SSI, where whole genome sequencing (WGS) of the STEC isolates began routinely in 2015. Selected isolates prior to 2015 have also been subjected to WGS. The sequences are available at http://www.ncbi.nlm.nih.gov/bioproject/1149523 (accessed on 8 October 2025). This project represents 2590 sequences of Danish Shiga toxin-producing *E. coli* (STEC) during the period from 2014 to February of 2024.

### 2.5. Case Definitions

Isolation of the same organism from a patient within a six-month period was considered a single episode. Thus, simultaneous double-infections with two STEC strains or two identical STEC isolates at intervals over six months were registered as two episodes. Travel was defined as visiting any country outside Denmark within a two-week period prior to onset of diarrhoea.

### 2.6. Pathotypes

In order to characterise cross-over (or hybrid) pathotypes, isolates were defined as STEC if positive for *stx1* and/or *stx2*, ExPEC if positive for two or more of *papAH* and/or *papC* (P fimbriae), *sfa/focDE* (S and F1C fimbriae), *afa/draBC* (Dr-binding adhesins), *iutA* (aerobactin siderophore system), and *kpsM* II (group 2 capsules) [16], and as UPEC if positive for two or more of *chuA* (heme uptake), *fyuA* (yersiniabactin siderophore system), *vat* (vacuolating toxin), and *yfcV* (adhesin) [17]. Enterotoxigenic *Escherichia coli* (ETEC) were defined as positive for heat-labile toxin (LT) genes, *elt*A and *elt*B and/or heat-stable enterotoxin (ST) genes, *est* as described in Scheutz et al. [18].

### 2.7. Update of the CGE VirulenceFinder Database and Virulence Profiles

The VirulenceFinder database at the Center for Genomic Epidemiology (CGE) [18,19,20] was expanded to include an improved detection of 68 *eae* alleles found in *E. coli* and *E. albertii*. Addition of genes found in *Shigella* and EIEC included *icsA*, *ipaD*, invasion plasmid antigen genes *ipaH*, *ipaH*7.8 (both plasmid encoded), *ipaH*9.8 (chromosomal) and *lacY*. WGS sequences were examined for the presence of the pathogenicity island Locus of Adhesion and Autoaggregation (LAA). Eight marker genes for the four modules on the 86 kb chromosomal mosaic element LAA described by Montero et al. [21] included *sisA*, *hes*, (module I), *iha*, *nmpC*, *lesP*, (module II), *pagC*, *tpsA*, (module III), and *ag43*, (module IV), see T2 in Appendix A (SM) for accession numbers and translation of previously used designations of *eae* alleles.

### 2.8. Surveillance Data

Through the laboratory notification system, STEC cases were notified to the Danish Registry of Enteric Bacteria and isolates were submitted on a voluntary basis to the NRL at SSI for characterisation. As of 14 April 2000, STEC infections and clinical HUS were also added to the list of individually notifiable diseases. The sequences are available at http://www.ncbi.nlm.nih.gov/bioproject/1149523 (accessed on 8 October 2025). This project represents 2590 sequences of Danish Shiga toxin-producing *E. coli* (STEC) during the period between 2014 to February of 2024.

### 2.9. Clinical Information

Basic clinical information was obtained from submitting laboratories, the National Gastrointestinal Case Notification Register (TBR), and practitioners’ paper forms for all cases. From 1997 to 2005, 562 patients were interviewed using a standardised questionnaire including clinical information and possible exposures [22,23]. Medical characteristics and symptoms among Danish sporadic culture-confirmed STEC cases (176 adults and 120 children) from a case–control study from 2018–2020 were also included in the data [24]. An additional 21 patients were interviewed using a similar questionnaire in 2007. Finally, medical records were obtained from 11 patients with HUS.

### 2.10. Statistical Methods

Background population data (1997–2023) were obtained from Statistics Denmark, Danmarks Statistik.

### 2.11. Ethics

This study was performed as a national disease surveillance activity under the auspices of the Statens Serum Institut as per the Danish Health Act § 222. According to Danish law, ethical approval is not required in the surveillance of STEC for the combination of microbiological data with gender, age, clinical outcome, hospitalisation, travel history, and seasonality. Individual consent was obtained from all interviewed patients and/or their parents.

## 3. Results

### 3.1. Incidence

A total of 8038 cases of STEC were identified during the study period (1997–2023). An increase in incidence was observed over time from 0.6 cases in 1997 to 24.0 cases in 2023 per 100,000 people, except during the COVID-19 pandemic years 2020 and 2021, Figure 1. Notably, case counts in 2022 and 2023 more than tripled compared with pre-pandemic levels, peaking at 1434 cases in 2023. The highest incidences were observed in 2023 among children aged <1 year and 1–4 years, with 95.2 and 69.4 cases per 100,000 children, respectively, Figure 1. Incidence by age less than five and more than five years is specified in T4 in Appendix A. There were 5156 confirmed isolates. A total of 977 were notified without the submission of an isolate, and 1687 submitted isolates could not be confirmed at SSI. A total of 218 DNA extracts were *stx*-positive by PCR with one to six extracts per year from 2003 to 2018, and 29 to 55 (average 40) per year from 2019 to 2023. Information on HUS was available for 7270 and missing for 768 cases. A total of 198 cases of HUS were registered. A total of 138 STEC isolates from 131 HUS cases were available for typing, Table 1.

### 3.2. Season, Age, and Sex

A typical seasonal pattern with a prolonged summer peak was observed. The incidence in small children had the most pronounced seasonality compared to other age groups (Figure 2). STEC isolates carrying the *eae* gene had stronger seasonal variation compared to *eae*-negative isolates (Figure 3). The (late-) summer peak of the *eae* positive STEC can mainly be explained by domestically acquired infection by children aged 1–4 years with an average of 40.9 (range 29–51) cases per month during winter (November to May) and an average of 91.6 (range 73–127) cases per month during the (late-) summer (June to October), T3 in Appendix A. A total of 46 to 50 percent of patients less than 20 years old and 57% to 60% of patients aged 19 to 49 and more than 69 years old were females, as were 62–64% between 50 and 69 years, T4 in Appendix A.

### 3.3. Prevalent Serotypes

The most common O groups were O157, O103, and O26. The predominant H types in each O group are shown in Table 2. Thirty-four O groups with more than 20 isolates each accounted for 4394 (86%) of 5137 isolates with an O group. A total of 415 (8%) isolates of the same O group were isolated 6 to 19 times and belonged to 39 different O groups (Group 1). Three percent (176 isolates) representing 1 to 5 isolates of the same O group belonged to 76 different O groups (Group 2). Finally, 142 (3%) were O rough, Table 2.

### 3.4. Travel

In the study data, 3302 (64%) patients had acquired their STEC infection in Denmark, while 923 (18%) reported foreign travel. Information on travel was not available for 912 (18%) patients, Table 2. Europe followed by the Middle East and Africa were the most visited destinations for travellers. Of the travel-related strains, 40% (368/920) were *eae*-positive compared to 64% (2095/3295) of domestically acquired strains. Half (439 = 48%) of the travel-related strains could be typed for their O group and for presence of the *eae* gene and belonged to five O groups: *eae*-negative O117 (14%), *eae*-positive O157 (14%) and O26 (8%), and *eae*-negative O146 (6%) and O128 (6%). A total of 41% of travel-related O157 infections were acquired in Central Europe (Bulgaria, 5%) or Mediterranean countries (Turkey, 35%; Egypt, 7%; Spain, 11%; Greece, 5%; and Italy, 4%). Travel-related infections primarily acquired outside Europe were of the O groups O186 (9/11; 82%), O117 (129/180; 72%), and O181 (13/23; 57%). O117 isolates were isolated each year (1–16 isolates per year throughout the study period) and were related to many different regions, primarily Africa (Egypt: 15 cases), Asia (India: 15 cases) and the Middle East (27 cases). Twenty-one infections were O104 (21/34; 62%) associated with the German outbreak of O104:H4 in 2011 [9]. The proportion of domestically acquired O groups among the 10 most common O groups were *eae*-positive O26 (72%), O63 (66%), O103 (77%), O111 (53%), O121 (92%), O145 (74%), and O157 (73%), and *eae*-negative O27 (62%), O91 (59%), and O146 (64%).

### 3.5. Haemolytic Uraemic Syndrome (HUS) and Bloody Diarrhoea

Sixty-three out of one-hundred-and-twenty-four (51%) HUS cases with only one STEC isolate were preceded by bloody diarrhoea and nine (7%) by non-bloody diarrhoea, and two were from urinary tract infections as previously described [25]. Bloody diarrhoea was registered in a total of 1749 cases, including 676 with and 1073 without bloody stools, and was higher in *eae*-positive cases, 676 (46%), than in *eae*-negative cases, 161 (25%). In 1114 *eae*-positive STEC cases, bloody diarrhoea was registered in 127 (11%) children 1–4 years old, in 25 (2%) children less than 1 year old, and in 44 (4%) children 5–9 years old, Figure 4. *stx1a*, *stx2a*, and *stx2c* alone or in combination accounted for 485/515 (94%) of cases with bloody diarrhoea. In 635 *eae*-negative STEC cases, bloody diarrhoea was registered in 20 (0.9%) cases in children less than 10 years old and in 141 (22.2%) teenagers and adults, Figure 4. *stx2b*, *stx2a*, and *stx1c* alone or in combination accounted for 137/161 (85%) of cases with bloody diarrhoea.

In 131 HUS cases, the submitted culture was confirmed as STEC. Information on *eae* and *stx* subtyping from patients with one STEC type, and where HUS was registered as either present (121 cases) or not present (4369 cases), was obtained for 4490 isolates, Table 3**.** A total of 98 of 108 (91%) *eae*-positive HUS cases were domestically acquired with *stx2a* plus/minus *stx1a* and/or *stx2c* and 67 cases were in children less than 10 years old, Figure 5a. Half, 6/13 (46%), of the 13 *eae*-negative HUS cases were *stx2a* from adults with a history of travel, Figure 5b.

Four out of seven HUS cases with double infections (see below) were preceded by bloody diarrhoea, T5 in Appendix A. This information was not available for the remaining three HUS cases. Thus, information on *eae* and *stx* subtyping was available for 138 isolates from 128 cases of HUS. The seven HUS patients with double STEC infections were of the serotypes O21:H8/O145:H^−^, O26:H11/O174:H21, O55:H7/O121:H19, O101:H1/O104;H4, O103:H2/O157:H7, and two cases of O145:H^−^/O157:H^−^. From 121 patients with one STEC isolate, which could be subtyped for *stx* and *eae*, O157:[H7] was isolated from 47 (39%) HUS cases, and non-O157 from 74 (61%): O26:[H11] from 23 cases, O145:[H28] (12 cases), O104:H4 and O111:[H8] (8 cases each), O121:[H19] (6 cases), O103:[H2] (4 cases), O183:H18 (2 cases), and O8:H19, O13/O73:K1:H18, O55:H12, O85:H4, O104:H7, O137:H6, O165:K-:H-, O168:H8, O174,H2, O177:H11, O177:H25, and O182:H21 (1 case each). *stx2a* alone or together with *stx1a* and/or *stx2c* was found in 99/108 (92%) *eae*-positive and 10/13 (77%) *eae*-negative STEC isolates. The 12 cases with subtypes different from the above are described in the footnotes to Table 3.

Seventy-four (56%) HUS patients with culture-confirmed STEC were 5 years old or younger, of which 56% were females. Fourteen (11%) were 5–9 years old, of which 86% were females. A total of 17 (13%) were more than 60 years old, of which 53% were females, and 26 (20%) were between 10 and 59 years old, of which 54% were females. Six deaths, where STEC was isolated, were recorded: four *eae*-positive (one O128:H-, *stx2f*, and three O157:H7, *stx1a* and *stx2a*), and two *eae* negative (one O17:H41, *stx2c*, and one O rough:H30, *stx2*). A non-STEC *eae*-positive O157:H- was isolated from one lethal case of HUS and one death with HUS was notified but no isolate was submitted for typing.

Seven of twenty-five (32%) patients with culture-confirmed EAEC-STEC O104:H4 (*stx2a*) developed HUS and were part of the German outbreak in 2011 and are described elsewhere [9,26].

### 3.6. Multiple Infections and Long Term Carriers of STEC

Double infections with two different STEC isolates were registered in 59 patients. In addition, one patient was infected by three STEC isolates. The most common serotypes were O157:[H7] (15 patients), O26:[H11] (10 patients), O103:[H2] (12 patients), O145:H-/H28/H34 (11 patients), and O121:H19 (4 patients). These five serotypes were found in different combination with each other in 14 of the patients with double infections. The combinations of *stx* subtype(s) and *eae* are shown in T5 in Appendix A.

Twenty-one patients were long term carriers with various degrees of gastrointestinal symptoms and with the same STEC for more than 6 months. These patients were infected with O rough:[H16], *stx1a* (2), O2:[K1]:H6, *stx2b* (4), O113:H4, *stx1c* or *stx2b* (1), O27:H30, *stx2b* (4), O55:H12, *stx1a* (1), O146:H28, *stx2b* (2), O146:H21, *stx2b* (3), O128:H2, *stx1a*; *stx2b* (1), O17/O77:H45, *stx1a* (1), O174:H8, *stx1c* (1), and O1:H20, *stx2a* (1). Three of these patients were colonised for more than a year: one for 7 years who was diagnosed with colitis ulcerosa (O2:K1:H6, *stx2b*) and two (O146:H21, *stx2b*, and O27:H30, *stx2b*) for three years.

### 3.7. Pathotypes

WGS on 2678 isolates was used to determine cross-over (or hybrid) pathotypes, T7 and T8 in Appendix A. The most common cross-over pathotypes were 194 STEC-UPEC, which are usually associated with urinary tract infections (UTIs) and/or bacteraemia. Out of 182 *eae*-alpha-positive pathotypes, 181 (90%) were *stx2f*-, *eae*-alpha-, *chuA*-, and *yfcV*-positive. A total of 157 of these were of ST583, but of different serotypes, all of which were of H type H6 (O63:H6, O71:H6, O125:H6 and O128:H6), followed by ST6044 [23] of serotype O63:H6. An additional 12 (26%) out of 46 *eae*-beta STEC-UPEC cases were *stx2f*, *eae*-beta, 10 of which were *chuA*-, *vat*-, *fyuA*-, and *yfcV*-positive strains. Among *eae*-negative cases, STEC-ExPEC/UPEC were most commonly found in *stx2b*-positive cases, appearing in 76 (16%) out of 468 cases. Twenty-six were also positive for *papAH/papC*, *sfa/focC*, *kpsMII* (=ExPEC), *chuA*, *fyuA*, *vat*, and *yfcV* (=UPEC), and 29 for *papAH/papC* and *chuA, fyuA, vat,* and *yfcV* (=UPEC). In 67 (44%) out of 157 *stx1a*-positive cases, 46 were *iutA* and *kpsMII* (=ExPEC)-positive, as well as *fyuA2* and *yfcV2* (=UPEC)-positive, STEC, see T7 in Appendix A.

STEC-ETEC pathotypes were found in 33/34 (97%) *stx2g*-positive strains. Twenty of these were of the serotype O187:H28, ST200, *estap*-positive, and seven were [O105]:[H7], ST5822. These and additional combinations are shown in T8 in Appendix A.

*lacY* (lactose permease) has been suggested as a gene that could be used to distinguish between *Shigella* and EIEC [27]. A total of 71 (2.7%) isolates were *lacY*-negative with 48 (68%) being of serotype O117:(K1):[H7], *stx1a*, *eae*-negative ST504 (25), ST5292 (17), ST6880 (3) and ST10599 (1). Among the remaining 25 isolates, 10 different serotypes were *stx2f*-positive, T9 in Appendix A.

### 3.8. Virulence Profiles

In 2633 WGS sequences, none of the LAA marker genes were found in 129 *eae*-negative and 393 *eae*-positive sequences. The LAA module marker genes *sisA* and *hes* (module I) and *nmpC* (module II) were found exclusively in, respectively, 527 (50%), 125 (12%), and 818 (78%) out of 1047 *eae*-negative strains, whereas *pagC* (module III) with or without *iha* (module I) was almost exclusively found in 662/1064 (62%) *eae*-positive strains and only in 103/1047 (10%) *eae*-negative strains. *iha* was present in 872 (83%) *eae*-negative and 722 (68%) *eae*-positive strains, *lesP* in 357 (34%) *eae*-negative and 44 (4%) *eae*-positive, *tspA* in 335 (32%) *eae*-negative and 218 (20%) *eae*-positive, and *ag43* in 373 (32%) *eae*-negative and 459 (20%) *eae*-positive strains. All eight LAA marker genes, *sisA* and *hes* (module I), *iha*, *nmpC* and *lesP* (module II), *pagC* and *tpsA* (module III), and *ag43* (module IV), were found in 23 *eae*-negative strains, which were *stx1a*, O91:H14 (3), *stx1a* or *stx2d*, O96:H19 (1), O174:H2 (1), *stx2a*, O89:H11 (1), O113:H21 (3), O174:H2 (1), *stx2b* + *stx2c* + *stx2d* O153:H25 (1), *stx2c* O8:H21 (1), and one each of *stx2d* O91:H21, O113:H21, O163:H19, and O174:H21. *iha* and *ag43* in different combinations were found in 859/908 (95%) *eae*-positive strains. The combinations of the eight LAA marker genes are shown in Figure 6.

The number of diagnostic results reported to the Danish Registry of Enteric Bacteria that included more than one DEC type from the same case within a six-month period increased from 7 in 2020 to 296 in 2023. ETEC-*ipaH*-EIEC had the highest increase from 1 case in 2015 to 143 in 2023, followed by STEC-ETEC with 1 in 2015 to 125 in 2023. A total of 908 cases were reported to have two or three DEC types diagnosed, T10 in Appendix A.

### 3.9. Outbreaks and Clusters

Several outbreaks were detected during the study period and have previously been described [5,6,7,9]. Approximately 230 small clusters with between two to five isolates were typically registered per year during 2015–2023.

## 4. Discussion

The number of STEC cases has increased steadily since the implementation, in 1997, of methods of molecular detection of STEC in Danish patients. During the first six years, molecular detection was exclusively performed at SSI, covering approximately half of the country. By 2003, regional hospitals began using PCR, and an increase in STEC coincided with the increasing number of primary diagnostic laboratories at regional hospitals implementing the guidelines of the Danish Society for Clinical Microbiology. The initial guidelines recommended molecular detection methods and that all children less than 7 years old and all patients with bloody diarrhoea or a travel history should be examined for STEC (and other DEC). Varying principles and methods used for testing in primary diagnostics therefore influenced the detected incidence over time and across regions. The introduction of molecular detection methodologies has resulted in a change in selection criteria so that almost all diarrhoeal stools from 2021 are now being examined for DEC, including STEC.

The increase may therefore mainly reflect changes in methodology, clinical selection criteria, and awareness rather than a true increase. A similar increase in Norway was also mainly explained by increased awareness and changes in methodology [28]. Changes in testing practices have been used to explain the increase in STEC incidence, especially of non-O157 cases, in the US [29,30], and in Australia, it has been suggested that a slight increase in the incidence of STEC may be linked to an increase in the number of stool samples being tested and changes in laboratory methods used to detect STEC [31]. The true incidence can only be estimated if all patients with diarrhoea are examined for STEC. The observed incidence of 24 STEC cases per 100,000 people in Denmark is the highest country-specific notification rate in the EU, followed by Ireland, Malta, and Sweden (15.8, 12.2, and 8.9 cases per 100,000 people, respectively), and STEC is now the third most reported zoonotic agent in humans [4]. Based on our present findings, we strongly recommend that all faecal specimens should be examined for STEC.

The incidence in less-than-one-year-old children rose to 95.2 in 2023 and to 69.4 in 1–4-year-olds. This confirms that children less than 5 years old are especially vulnerable to STEC infection.

One-hundred-and-forty-nine different O groups were identified, with O157 (16%) being the most prevalent followed by O103 (11%) and O26 (10%). O145 and O111, which are often included in what has been termed “the gang of five”, i.e., O26, O103, O111, O145, and O157, accounted for 5% and 2%, respectively. Thus, “the gang of five” were only found in 44% of cases. O146 (8%), O128 (4%), O91 (4%), and O27, O63, and O117 (3% each) together were found in 25%. O group O104 and O117 isolates were primarily found in travellers. O104 isolates were related to the German outbreak in 2011 where patients had been in Hamburg or elsewhere in Germany [26]. O117 isolates were primarily related to travel to Africa, Asia, and the Middle East—see Results. This is in line with a study in the UK where 19 isolates originated from the same regions [32]. Frontline diagnostic microbiology laboratories in England and Wales originally misidentified these strains as *Shigella sonnei* or *Shigella* spp., probably because of the unusual biochemical phenotype exhibited by this serotype [32]. It is typically lactose-negative. Consistent with this, we found all 48 O117 isolates sequenced by WGS negative for *lacY*. No reports from these regions have to our knowledge described findings of O117:K1:H7 and it is plausible that this pathotype is misidentified as *Shigella* and thus overlooked. We have previously described how it is associated with persistent diarrhoea in Danish patients [33].

Fourteen O groups (O26, O27, O55, O63, O91, O103, O111, O121, O128, O145, O146, O156, O157, and O177) were found in 74% of patients without a travel history, indicating that these O groups are found in yet unknown Danish reservoirs or foods. More than 550 different O(:K):H serotypes were found. Our finding that non-O157 O groups were by far more common than O157 is in line with other European studies [34,35,36,37,38,39,40] and recent EU reports [3,4] and different from studies in the USA, where O157 was found in one-half of cases [29,41,42,43], and in Argentina, where O157 was found in 58.8% of cases [44]. It is quite evident that different serotypes and clones vary from region to region [45], which in turn lends support to the theory of local clonal spread of particular types of STEC [46,47].

It has been suggested that not all *stx* subtypes have been isolated from human diseases [48]. In this study, all 3 *stx1* subtypes and 11 out of 15 *stx2* subtypes were found in patients with clinical symptoms. Seventeen *stx* subtype profiles were found in seven or more isolates. *stx1a* was by far the most common *stx* subtype either alone in or in combination with other *stx2* subtypes. *stx2b* was the second most common subtype and found almost exclusively in *eae*-negative isolates either alone or with *stx1a* or with *stx1c*. Sixteen patients with *stx2b* in *eae*-negative strains had carried them for more than a year. *stx2b* was followed by *stx2a* alone, primarily in *eae*-positive and in *aggR*-positive strains and in combination with *stx1a* with *stx2c* or both. The *stx1d* subtype in *eae*-negative isolates has been indicated as significantly associated with vomiting [49], whereas patients with subtypes *stx2e* and *stx2g* seem to have milder symptoms (our data are not complete). *stx2f* was primarily found in *eae*-positive strains. The high number of *stx* profiles is most probably a result of *E. coli* strains being particularly susceptible to infection by multiple types of the bacteriophages that encode Shiga toxin but also a result of Shiga toxin in and of itself being capable of causing disease in humans. The high degree of diversity in serotypes and *stx* profiles may well be a more complete representation of the actual repertoire of STEC types and reflects that detection has been directed at the *stx* genes rather than towards specific O groups. A similar diversity has been seen in countries using methods targeting the *stx* genes [28,50,51]. The observed seasonality in *eae*-positive STEC, primarily with *stx1a* subtypes (and/or *stx2a* or *stx2c*) having summer peaks, may in part be explained by the presence of these subtypes in *eae*-positive O groups such as O26, O103, and O157, which all exhibit similar summer peaks. The (late-) summer peak of *eae*-positive STEC also coincides with domestically acquired infection by children aged 1–4 years, with a lower number of cases during winter and a higher number of cases during the (late-) summer, T3 in Appendix A. *stx2b*—either alone or together with *stx1c*—does not exhibit a summer peak. Both *stx1c* and *stx2b* are almost exclusively found in *eae*-negative STEC strains, and *stx1c* +/− *stx2b* are evenly distributed over the entire year without a summer peak. To our knowledge, this has not been described before and could indicate differences in reservoirs, transmission, or *stx* bacteriophage ecology.

Sixty patients were found positive for two different STEC types, highlighting the importance of examining several colonies from infected patients. O157 was found in 15 of 60 cases with double infection, T5 in Appendix A. Diagnostics directed towards O157, such as sorbitol McConkey (SMAC) plates or SMAC plates supplemented with cefixime and tellurite (CT-SMAC), are likely to focus only on sorbitol-negative colonies and thus likely to overlook other types of STEC in double infections. Double infections are not submitted to the European surveillance database TESSy at the European Centre for Disease Prevention and Control (ECDC) and may therefore be underreported in the EU. Double or triple infections were described in a UK Health Security Agency (UKHSA) report published in 2025, where 26 out of 2063 culture-confirmed cases of STEC were co-infections and the most commonly reported serogroups were O157 (n = 7), O146 (n = 6), and O26 (n = 5) [52]. The same *stx* subtype was found in two strains from the same patient in sixteen cases where *stx1a*, *stx2a*, *stx2f*, and *stx2b* were found, T5 in Appendix A. This could represent cross-infection with identical bacteriophages and warrants further studies. The increase in culture-independent diagnostic tests pose a threat to proper diagnostics of double infections and in risk assessment of each individual patient and may also endanger outbreak detection. It also complicates the interpretation and possible association with clinical significance of specific STEC types.

Young age (less than five years) and *stx2a* were associated with a higher risk of HUS. On a smaller dataset, we have previously shown that young age and presence of the *eae* and *stx2* genes are significant risk factors for HUS [53] and related to the *stx2a* subtype [54]. In a study in the UK, isolates from all cases of confirmed STEC-HUS encoded *stx2* (139 *stx2* and 23 *stx1* and *stx2*) [55]. This is the first study where 4605 different STEC isolates have been analysed for their *stx* subtypes, the presence of the *eae* gene, and the association to HUS regardless of their serotype. In this larger analysis, only young age and *stx2a* were associated with an increased risk for HUS. Similar high-risk factors for HUS in Norway from 1992 to 2012 were age ≤ 5 years (OR: 12.2) and *stx2a* (OR: 92.7) [28]. This association between young age and presence of STEC with the *stx2a* gene has been demonstrated in several studies [50,56,57]. Surveillance reports within the European Union (EU) member states and European Economic Area (EEA) countries have consistently reported young age as a risk factor for HUS [58,59,60,61,62,63,64] and found an increased risk of HUS associated with *stx2a* or *stx2d* [65]. *stx2d* has been suggested as an indicator for severe clinical outcomes such as HUS or bloody diarrhoea [66] and is also indicated as having higher predictive value for the potential to induce HUS in a recent assessment by the French Agency for Food Safety, ANSES [67]. This was not indicated in our analysis even though 3 out of 69 patients with *stx2d* positive STEC isolates developed HUS and 1 patient out of 25 with *stx1a* and *stx2d* (O183:H18) developed HUS. Even though the number of isolates with *stx2d* in our study was low, international reports indicating *stx2d* as a predictor for HUS [65,67] have led to its inclusion in the definition of HUSEC in Denmark. Bloody diarrhoea incidence was higher in *eae*-positive cases and primarily registered in children, Figure 4. Specific subtypes in *eae*-positive STEC, *stx1a*, *stx2a*, and *stx2c* alone or in combination, accounted for 94% of cases with bloody diarrhoea, and were domestically acquired. *eae*-negative STEC, with *stx2b*, *stx2a*, *stx1a*, and *stx1c* alone or in combination, accounted for 85% of cases with bloody diarrhoea, and only 40% were domestically acquired. It is therefore relevant to consider prioritisation of detection and surveillance on patients with domestically acquired bloody diarrhoea and HUS.

Ten patients infected with non-*stx2a* or non-*stx2d*, i.e., low-risk STEC isolates, developed HUS. One patient was part of a small outbreak where other STEC strains were isolated. Considering that we have found double infections in 60 Danish patients, it cannot be excluded that a HUS-associated strain was overlooked in this patient during the outbreak. Three patients developed HUS and were treated with various antibiotics during the acute phase of disease. Both in vitro and clinical studies have suggested that DNA synthesis inhibitors should not be used to treat patients with STEC infection [68]. The antibiotics used in the three cases were all different. One patient with O104:H7 (*stx1c*) was hospitalised with nephrotic syndrome, illustrating that all cases need to be assessed individually. The last patient with O157:H7 (*stx2c*) was kindly determined by Tim Dallman to belong to lineage I/II, clade 8, a highly pathogenic clade of O157 [30,44] known to have both *stx2a* and *stx2c* genes. None of the 59 O157 strains in a Japanese study with *stx2c* alone belonged to clade 8 [69], and it is therefore possible that a *stx2a*-containing bacteriophage was lost during the course of disease and/or during culture of the isolate from this patient. Such loss of *stx* genes has been reported in cases of HUS [70] and haemorrhagic colitis [71], upon sub-cultivation [72], and from asymptomatic long term shedders [73]. The *stx2a* subtype is not always found on identical bacteriophages in O157 strains [69] and it will be interesting to study if this is also the case in non-O157 strains and for other subtypes of *stx*.

WGS was performed on 2678 isolates from 2620 patients out of a total of 4605 isolates. Out of 201 cross-over pathotypes, 194 *eae*-positive isolates would be characterised as STEC-UPEC and 7 as STEC-ExPEC. For the *eae*-negative isolates, 47 could be characterised as STEC-UPEC, 98 as STEC-ExPEC-UPEC, and 25 as ExPEC. Thus, a total 369 (14%) would potentially be able to cause either urinary tract infection (UTI) and/or bacteraemia. Of particular interest was the fact that two-thirds (194/296, 65.5%) of *stx2f eae*-positive isolates were STEC-UPEC. *stx2* was detected in 9/193 (4.7%) blood isolates in a Norwegian study of isolates from adult bacteraemic patients [74]. An examination of 168 faecal isolates submitted to the Norwegian Institute of Public Health (NIPH) from clinical microbiology laboratories across Norway found that 10 out of 31 STEC isolates carried ExPEC-related genes and concluded that hybrid ExPEC-IPEC (intestinal pathogenic *E*. *coli*) strains were found at a very high frequency [75]. STEC O128:H2, *stx1c*, *stx2b* was isolated from a patient with bacteraemia and HUS [76]; O78:HNM, *stx1c*, *hlyA* in blood and faecal samples of a 2-week-old boy who had bacteraemia and HUS [77]; and STEC O157:H7 from a urinary tract infection complicated by bacteraemia and HUS that was not preceded by diarrhoea (D-HUS) [78]. Two Danish cases of HUS with urinary tract infection (UTI) and without a previous history of diarrhoea have been described [25]. Unexpected childhood death due to haemolytic uremic syndrome was reported in a 21-month-old girl infected with O157 and from a 4-year-old girl with culture from urine and faeces that grew STEC [79]. As characterisation of isolates from UTI and bacteraemia is not routinely performed in Denmark, the prevalence of these types is unknown. Furthermore, 51 isolates were STEC-ETEC, dominated by two-thirds (33/51 64.7%) *stx2g eae*-negative strains. LAA was originally described as being exclusively present in a subset of emerging LEE-negative STEC strains [21]. We only found 23 out of 893 *eae*-negative isolates positive for the full set of the eight marker genes for LAA, also called the “complete” structure [80], Figure 6. Specific and different combinations were almost exclusively found in *eae*-positive isolates (*iha* and/or *agg43*). This is in contrast to a previous study that did not find LAA marker genes, either “complete” or “incomplete” (<4 modules), in LEE, i.e., *eae*-positive isolates [80]. In our study, different combinations were almost exclusively found in *eae*-negative isolates. In particular, *nmpC*, often together with *iha*, was only found in 820/1050 (78%) *eae*-negative isolates, Figure 6.

Denmark has seen an increase in the incidence of STEC over the past 27 years, which is most probably due to an increase in the number of examined patients and changes in methodology, in particular syndromic test panels and multiplex PCR on all patients. The increase in the number of cases of enteric pathogens was already reported in Denmark in 2018, when it was concluded that PCR-based test methods lead to increased detection rates [81]. The increase in STEC was mainly due to an increase in the number of notifications of STEC cases without an isolate, Table 1. Since 2018, the number of isolates submitted for characterisation has remained relatively stable, around 300–400 per year. The introduction of syndromic test panels and multiplex (Q-)PCR for all patients has also led to an increase in double and triple DEC infections, similar to what is seen with other diarrhoeagenic pathogens reported to the Danish Enteric Bacterial Registry. As isolates are now less frequently obtained from such specimens, the role of clinical microbiologists in interpreting and assessing the significance of diagnostic results has become increasingly important. Double infections and outbreaks may go undetected, making it increasingly difficult to identify sources and routes of transmission. Significant risk factors for HUS are young age and subtype *stx2a*, regardless of serotype or other virulence genes. As of September 2015, these findings have led to a change in the clinical management of STEC infections in Denmark. Patients infected with STEC strains carrying *stx2a* or *stx2d*—so-called HUS-associated *E. coli* (HUSEC)—are considered to be at a higher risk for developing HUS and for transmitting STEC to close contacts. These patients are quarantined from institutions (e.g., day cares, nurseries, and kindergartens) and from occupations involving food handling. Restrictions are also applied in hospitals, clinics, and nursing homes until two consecutive stool samples test negative for STEC carrying *stx2a* or *stx2d*. In contrast all non-*stx2a* or non-*stx2d* patients are considered low-risk for HUS and may return to institutions or work once gastrointestinal symptoms have resolved, regardless of their carrier status. Antibiotic treatment with protein synthesis inhibitors, such as azithromycin, may be considered in selected cases, when specific criteria related to patient group, serotype, virulence profile, and disease duration are met—particularly in long term carriers with persistent symptoms, as suggested by Agger et al. [68]. Finally, WGS revealed that 19% of isolates were cross-over pathotypes, highlighting the need for full characterisation of all STEC isolates.

## 5. Conclusions

Changes in diagnostic methodology have resulted in a significant increase in the number of diagnosed STEC cases. On the other hand, culture of STEC is declining, which poses a challenge to the identification of multiple STEC infections and outbreaks. Syndromic (PCR) test panels have also resulted in an increase in the detection of multiple microorganisms, which in turn has increased the role of clinical microbiologists in interpreting and assessing the significance of diagnostic results. The need for high-quality curation of surveillance data is more pertinent than ever. WGS has demonstrated that cross-over pathotypes such as STEC-UPEC, STEC-ExPEC, and STEC-ETEC are commonly present in diarrhoeal stools. Young age (less than five years) and *stx2a* were associated with a higher risk of HUS.

## Figures and Tables

**Figure 1 microorganisms-13-02342-f001:**
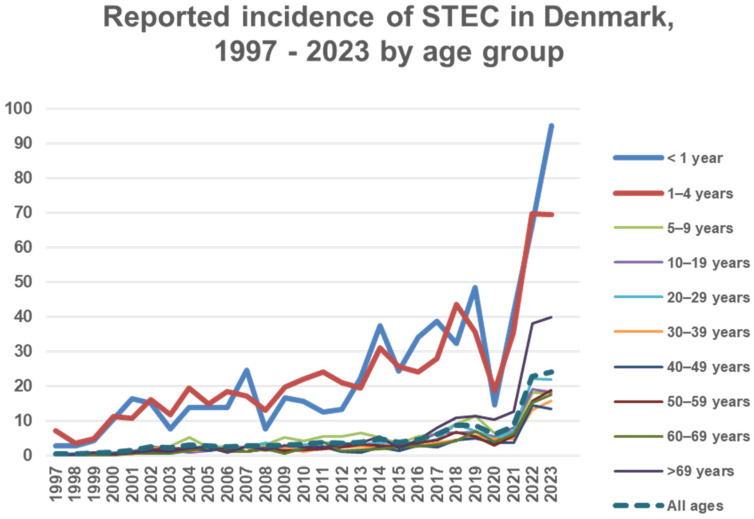
Incidence on the Y axis: number of STEC cases per 100,000 people in Denmark according to age group.

**Figure 2 microorganisms-13-02342-f002:**
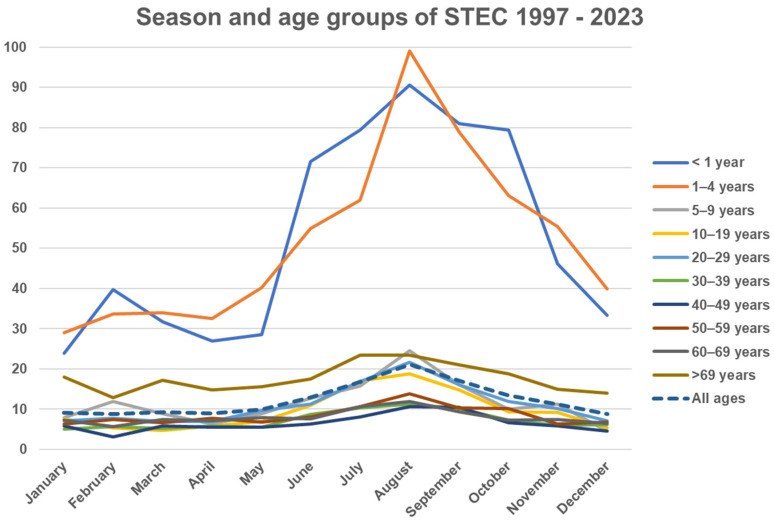
Y axis: Average monthly incidence of STEC per population of 100,000 per year in Denmark (1997–2023), stratified by age group.

**Figure 3 microorganisms-13-02342-f003:**
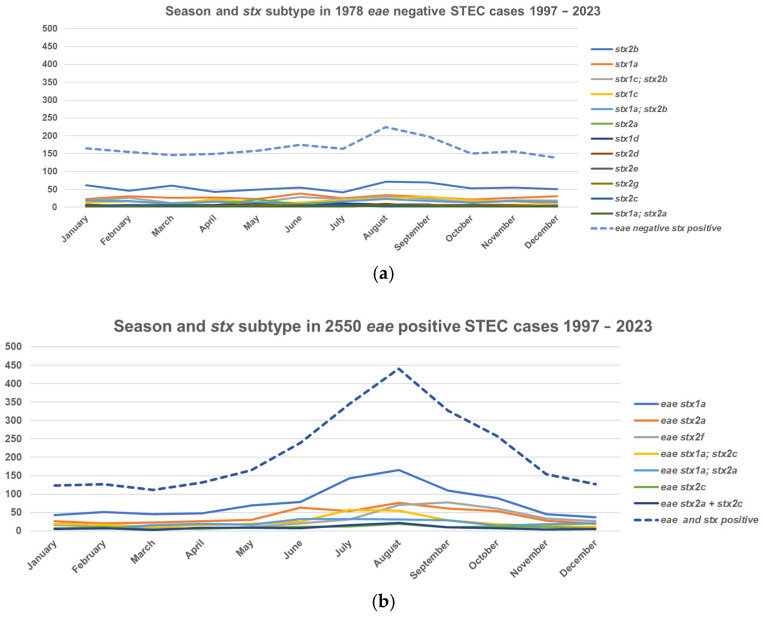
Seasonal variation and *stx* subtype in *eae*-negative and positive STEC in Denmark, 1997–2023. (**a**) Season and *stx* subtypes in 1978 *eae*-negative STEC cases. (**b**) Season and *stx* subtypes in 2550 *eae*-positive STEC cases. Only genes that were found in more than 20 isolates (range 20 *stx1a*; *stx2a*, *eae*-negative to 928 *stx1a*, *eae*-positive) isolates are shown.

**Figure 4 microorganisms-13-02342-f004:**
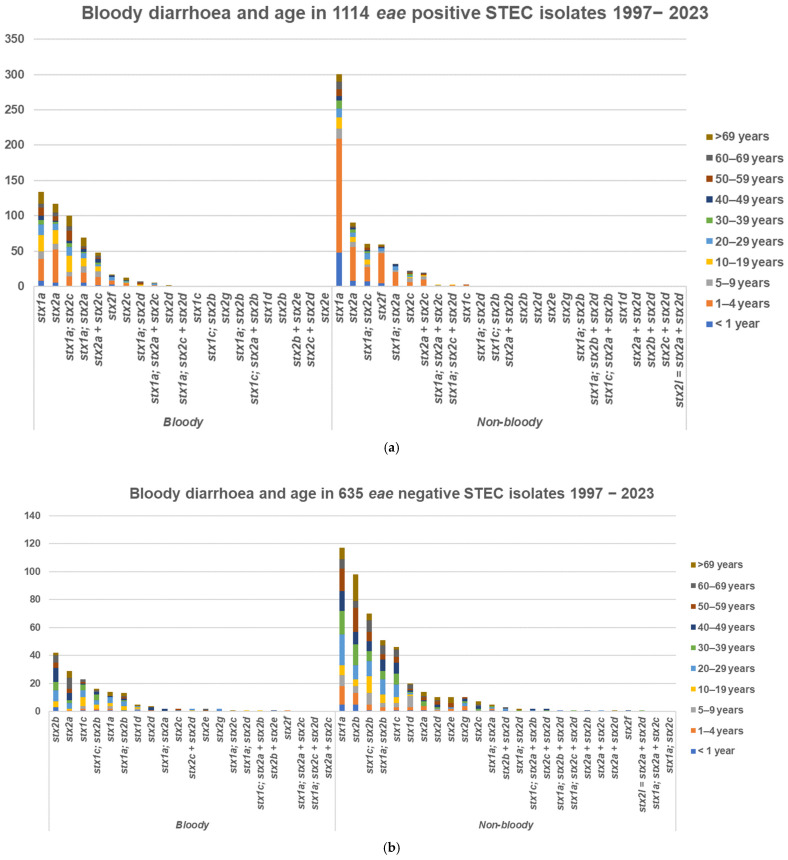
Bloody and non-bloody diarrhoea, 1997–2023. (**a**) In 1114 *eae*-positive. (**b**) In 635 *eae*-negative STEC cases.

**Figure 5 microorganisms-13-02342-f005:**
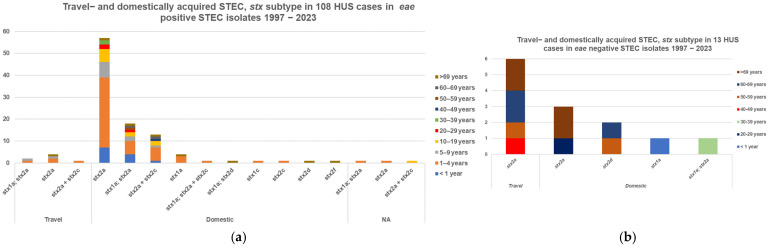
Number of HUS cases, *stx* subtype, and travel-acquired STEC in STEC in patients with only one STEC isolate; (**a**) 108 *eae*-positive cases, (**b**) 13 *eae*-negative cases.

**Figure 6 microorganisms-13-02342-f006:**
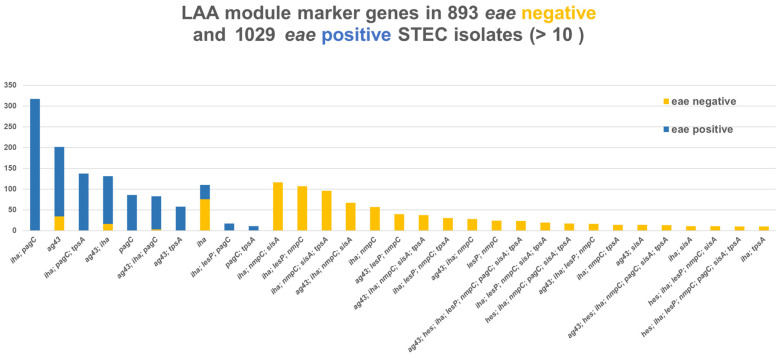
Combinations of eight LAA marker genes, *sisA* and *hes* (module I), *iha*, *nmpC* and *lesP* (module II), *pagC* and *tpsA* (module III), and *ag43* (module IV) in 893 *eae*-negative and 1029 *eae*-positive Danish STEC strains. Surveillance data.

**Table 1 microorganisms-13-02342-t001:** (**A**) Total number of STEC isolates available for typing and number of notifications from Danish patients with STEC infection during the period from 1997 to 2023. (**B**) The number of STEC isolates from HUS cases and number of notified HUS cases.

A				B		
Year	STEC Isolates	Notification of STEC Cases Without Isolate ^a^	Total Number of STEC Cases	STEC Isolates from HUS Cases	Notification of HUS Cases Without Isolate ^b^	Total Number of HUS Cases
1997	31	0	31	5	0	5
1998	29	3	32	3	1	4
1999	48	2	50	4	0	4
2000	64	1	65	4	0	4
2001	88	2	90	7 *	1	7
2002	139	4	143	1	1	2
2003	125	2	127	3	1	4
2004	152	16	168	6	0	6
2005	146	10	156	5 *	2	6
2006	135	14	149		3	3
2007	153	5	158	2	0	2
2008	155	3	158	5	0	5
2009	148	21	169	2	5	7
2010	167	18	185	3	1	4
2011	205	18	223	13 *	3	15
2012	185	22	207	8 *	3	10
2013	183	31	214	7	4	11
2014	229	52	281	6	4	10
2015	171	55	226	1	8	9
2016	201	67	268	6 *	1	6
2017	258	87	345	6 *	4	9
2018	304	188	492	11	8	19
2019	380	120	500	4	5	9
2020	239	91	330	2	2	4
2021	414	88	502	2	3	5
2022	398	939	1337	9 *	2	10
2023	409	1023	1432	13	5	18
Total	5156	2882	8038	138 *	67	198

In 21 cases, identical isolates were found in the same patient for more than 6 months and not included in the table. ^a^ Either the notification was submitted without an isolate (977), the submitted isolate could not be confirmed as STEC (1687), or the submitted DNA could only be confirmed by PCR (218). ^b^ Either the notification was submitted without an isolate (46), the submitted isolate could not be confirmed as STEC (18) or the submitted DNA could only be confirmed by PCR (3). * Double infection with two different STEC types were found in seven cases of HUS resulting in 138 isolates from 131 HUS cases.

**Table 2 microorganisms-13-02342-t002:** Prevalent O groups and H types in Danish STEC patients, 1997–2023.

O Group	Common H Type(s) ^(1)^	Number of Isolates	Percent	Number of Travellers	Percent Travellers Within this O group
O157	H7	817	16%	129	16%
O103	H2	550	11%	41	7%
O26	H11 (H46)	524	10%	71	14%
O146	H21, H28	405	8%	57	14%
O145	H34, H28 (H46)	251	5%	20	8%
O128	H2	209	4%	53	25%
O91	H14 (H21, H10)	191	4%	31	16%
O117	H7 (H8)	180	3%	129	72%
O63	H6	155	3%	12	8%
O27	H30	146	3%	19	13%
O111	H8	116	2%	36	31%
O121	H19	71	1%	3	4%
O113	H4, H6, H21	71	1%	18	25%
O174	H8, H21 (H2)	58	1%	14	24%
O8	H9, H19	57	1%	13	23%
O76	H19	53	1%	19	36%
O125	H6	47	1%		0%
O55	H7, H12	41	1%	4	10%
O177	H11, H25	40	1%	3	8%
O187	H28	36	1%	6	17%
O156	H25, H7	36	1%	15	42%
O2	H6, H4	36	1%	2	6%
O104	H4, H7	34	1%	26	76%
O54	H45	34	1%	2	6%
O5	H9	34	1%	6	18%
O80	H2	33	1%	2	6%
O166	H28	27	1%	5	19%
O154	H31	25	0.5%	2	8%
O40	H8	24	0.5%	7	29%
O78	H4	24	0.5%	6	25%
O132	H34 (H18)	24	0.5%	2	8%
O181	H16	23	0.4%	13	57%
O21	H21	22	0.4%	3	14%
O rough	H2, H7	142	3%	26	18%
Group 1 (39)		415	8%	95	23%
Group 2 (76)		186	4%	31	18%
Total		5137	100%	923	18%

^(1)^ Indicate the second or third most common H types.

**Table 3 microorganisms-13-02342-t003:** Virulence profiles found in patients with only one STEC type. Only the most common combinations of *eae*, *stx1*-, and *stx2* genes and subtypes are shown in this table. Eleven rare combinations and four single subtypes found in 30 patients without HUS are shown in the Appendix A T6. Virulence profiles found in patients with two different STEC types are shown in Appendix A T5.

	*eae*-Positive			*eae*-Negative			Total
*stx* Genes Subtypes	Non-HUS (n)	HUS (n)	HUS Attack Rate (%)	Non-HUS (n)	HUS (n)	HUS Attack Rate (%)	
***stx1* total**	**912**	**5**	**0.6%**	**586**	**1**	**0.2%**	**1504**
*stx1a*	907	^(a)^ 4	0.5%	323	^(b)^ 1	0.3%	1235
*stx1c*	5	^(c)^ 1	16,7%	200	-	-	206
*sstx1d*		-	-	63	-	-	63
***stx2* total**	985	**80**	**7.5%**	**887**	**11**	**1.2%**	**1963**
*stx2a*	401	62	13.4%	71	9	11.3%	543
*stx2b*	4		-	620	-	-	624
*stx2c*	110	^(d)^ 2	0.9%	34	-	-	145
*stx2d*	11	^(e)^ 1	8.3%	53	^(f)^ 2	3.6%	67
*stx2e*	2		-	49	-	-	51
*stx2f*	366	^(g)^ 1	0.3%	13	-	-	380
*stx2g*	2		-	40	-	-	42
*stx2a + stx2c*	89	15	14.4%	2		-	106
*stx2c + stx2d*	-	-	-	5	-	-	5
***stx1 + stx2* total**	**527**	**23**	**4.2%**	**442**	**1**	**0.2%**	**993**
*stx1a; stx2a*	208	21	9.2%	17	1	5.6%	247
*stx1a; stx2b*	1	-	-	175	-	-	176
*stx1a; stx2c*	285	-	-	4	-	-	289
*stx1a; stx2d*	13	^(h)^ 1	7.1%	11	-	-	25
*stx1a; stx2a + stx2c*	12	1	7.7%	-		-	13
*stx1a; stx2c + stx2d*	5	-	-	1			6
*stx1c; stx2b*	3	-	-	234		-	237
**Total**	**2424**	**108**	**4.3%**	**1915**	**13**	**0.7%**	**4460**

n: Number of cases with clinical information on HUS (present or not); -: No cases. Twenty-three isolates prior to 1997 are included in this table. ^(a)^ Four patients with O103:H2 were as follows: Two patients were treated with antibiotics, one due to suspicion of a urinary tract infection and one in the acute phase of diarrhoea. One patient was part of an outbreak with two STEC O157 types and several non-STEC types. One patient was co-infected with O157:H7 *stx1a*, *stx2a*, and *eae*. ^(b)^ One patient with O55:H12 was initially treated with antibiotics due to lower respiratory tract infection. ^(c)^ One patient with O104:H7 was probably nosocomially infected when hospitalised with nephrotic syndrome; ^(d)^ One patient was infected with O157:H7 of lineage I/II, clade 8, and one O157:H7 with no further information. ^(e)^ One patient was infected with O177:H25. ^(f)^ One patient was infected with O13,O73:K1:H18 and one with O8:H19. ^(g)^ One patient had O85:H4, ST 642, and diarrhoea for several weeks with thrombotic microangiopathy, haemolysis, and high creatinine levels. ^(h)^ One patient was infected with O183:H18.

## Data Availability

The data presented in this study are openly available in NCBI repository under accession number PRJNA1149523.

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
