# Peer review of "Shiga Toxin-Producing E. coli (STEC) from Danish Patients, 1997–2023: Diagnostic Trends and Bacteriological Findings"

_microorganisms, 2025, doi:10.3390/microorganisms13102342_

Round 1
Reviewer 1 Report
Comments and Suggestions for Authors
The authors describe the STEC surveillance in the period 1997-2023 as done in Denmark. A significant increase was seen in these years, largely due to changes in diagnostic methodology. The paper gives a nice overview of and insight in the Danish STEC surveillance. My comments are mostly textual, as it contains repetitions or is sometimes difficult to read.
Main comments
- Introduction, lines 63-86. In these sentences methods are described, which can also be found in the Materials and Methods. I would expect here the background to why the authors made this paper/overview.
- The authors have results on WGS analyses. Did you detect any, larger, clusters or outbreaks? Maybe the authors can add some sentences on this to the Results. Furthermore, in the case there were, could it have skewed the importance of an O type or virulence type?
- Discussion, lines 410-412. Do the authors have all faecal specimens tested for STEC? Or only those of patients with symptoms? This is an important difference, as persons can also be carriers, and -in my opinion- these persons are not (or at least less) of interest for public health surveillance. In the previous sentences, the authors mention that it is needed for estimating incidence. If carriers or cases with chronic infections are also of interest, then it should be reported as prevalence. Please elaborate a little bit more on your recommendation.
- Discussion, lines 443-467. This paragraph consists of only results, except the last sentence. I would suggest to summarize only the results that are also discussed here, in stead of repeat all the numbers.
- Discussion, lines 517-526. Here, only results are repeated without any elaboration or discussion. Please consider removing or summarize and discuss.
Minor comments
- 1 Study Period and Population, lines 95-98. Please consider replacing “next 16 years” and “next eight years” with the specific years, so that the reader does not have to calculate.
- 8 Surveillance Data, lines 154-156. I think these sentences fit better in 2.4.
- 4 Travel, line 240. I found “921/923” difficult to understand. For 921 of 923 information on travel was available? Please consider rewriting.
- 5 HUS and Bloody Diarrhoea, lines 258/261. Please consider adding A and B, respecively, to “Figure 4”.
- 8 Virulence Profiles, lines 361-362. I don’t understand this sentence, please reformulate.
- Discussion, lines 426-425. In the Results, only continents are mentioned. I would expect this sentence in the Results.
- Discussion, lines 590-591. I think this sentence belongs to the previous paragraph?
- I think something went wrong with the numbering, see for example reference 24 (page 20) and 69 (page 22).
- Supplementary files – excel. The numbering of the sheets and the numbering of the Tables on the relevant sheet do not match. Please check.
Reviewer 2 Report
Comments and Suggestions for Authors
microorganisms-3893569-peer-review-v1
In my opinion this is interesting survey on 27 year period after introduction of systematic bimolecular survey of STEC in Denmark. Authors have collected the available data from the government agencies and hospitals and treated and systematic explored with objective to give a clear vision of the Shiga toxin producing E. coli in Denmark. In my opinion paper deserve attention form the Editor and can be a good example for monitoring of one of the most relevant pathogens - STEC.
In my opinion paper can be rather called "survey" then a simple “article", since is based on systematization and analyzing of existing data. However, this is a topic from the competence of the editor of the journal.
I would like to recommend current paper be accepted for publication; however, some adjustments will need to be taken into consideration.
As this is research (survey) paper, the abstract will be beneficial if can be enriched with a bit more specific data that was reported later into the manuscript.
Introduction is quite general and providing information regarding STEC reality in Europe and with focus on Danish reality. Authors have provided overview of the importance of STEC for the public health and dynamic in diagnostic Danish situation, including principle diagnostic approaches applied.
Please, for the figures, explain Y axes.
Presume that for figure 2 this is number of cases per 100,000
Figures can be presented a bit better. Authors can standardize them and put them in same format and proportions.
Authors have provided detailed information regarding obtained results, based on the governmental program applied in Denmark. It is very interesting to observe spread of STEC prevalence, looking from position of seasonal, years, habits, symptoms perspective. Moreover, different genetic markers were evaluated as well.
In my opinion, even if the applied analytical methods were presented in the supplementary material, maybe it would be beneficial if the references on what these applied methods were performed will be cited in the mine text. This will give better accessibility to reference literature and will help readers to use current paper as reference work. Even maybe include the provided supplementary file in the principle body of the manuscript.
Discussion starts with statement that STEC has increased. In fact, maybe just before implementation of the surveying program this was just hidden existence of the STEC and simply not recorded/reported. Maybe authors can consider comment on this point. Lter on the manuscript this was discussed, maybe it will be more appropriate to be stated in the beginning of the discussion section.
In the discussion part authors have compared results related to their survey with that obtained from different countries, showing relevance to the explored topic.
Please, references needs to be formatted according to the exigences from the Publisher and the Journal.
Round 2
Reviewer 1 Report
Comments and Suggestions for Authors
The authors have answered my questions and comments satisfactorily. I have only one new comment: there is a typo in line 102: 1015 should be 2015.